# Comprehensive Proteome and Acetyl-Proteome Atlas Reveals Hepatic Lipid Metabolism in Layer Hens with Fatty Liver Hemorrhagic Syndrome

**DOI:** 10.3390/ijms24108491

**Published:** 2023-05-09

**Authors:** Li Zhang, Enling Wang, Gang Peng, Yi Wang, Feiruo Huang

**Affiliations:** Department of Animal Nutrition and Feed Science, College of Animal Science and Technology, Huazhong Agricultural University, Wuhan 430070, China; zhangli302@webmail.hzau.edu.cn (L.Z.);

**Keywords:** acetyl-proteome, FLHS, layer hens, lipid metabolism, proteome

## Abstract

The feeding of high-energy and low-protein diets often induces fatty liver hemorrhagic syndrome (FLHS) in laying hens. However, the mechanism of hepatic fat accumulation in hens with FLHS remains uncertain. In this research, a comprehensive hepatic proteome and acetyl-proteome analysis was performed in both normal and FLHS-affected hens. The results indicated that the upregulated proteins were primarily associated with fat digestion and absorption, the biosynthesis of unsaturated fatty acids, and glycerophospholipid metabolism, while the downregulated proteins were mainly related to bile secretion and amino acid metabolism. Furthermore, the significant acetylated proteins were largely involved in ribosome and fatty acid degradation, and the PPAR signaling pathway, while the significant deacetylated proteins were related to valine, leucine, and isoleucine degradation in laying hens with FLHS. Overall, these results demonstrate that acetylation inhibits hepatic fatty acid oxidation and transport in hens with FLHS, and mainly exerts its effects by affecting protein activity rather than expression. This study provides new nutritional regulation options to alleviate FLHS in laying hens.

## 1. Introduction

In the later stages of laying, hens are susceptible to lipid deposition in the liver due to nutritional and environmental factors, which can lead to the development of FLHS [1]. FLHS seriously affects the economic efficiency of the industry, as it decreases production performance and egg production rate, and even increases the mortality of laying hens [2]. Studies have shown that laying hens with FLHS have a phenomenon of hepatic metabolic remodeling, including lipid accumulation and the blockage of lipid transport [3]. Therefore, targeted nutritional modulation of the liver metabolism in laying hens with FLHS could be a potentially effective method. 

In the continuous research on metabolic fatty liver disease, post-translational modification is a hot topic for researchers [4,5]. The acetylation modification plays a significant role in multiple metabolic processes. Abnormal elevations of acetyl coenzyme A, an important substrate for acetylation modification, have been detected in the blood of animals with hepatic lipid deposition [6]. In addition, numerous studies have shown that acetylation activity changes with the nutritional and feeding status in response to different physiological states [7,8]. Notably, a recent study examined the hepatic metabolic changes in the liver of pre-laying and laying hens by analyzing proteome and acetyl-proteome [9]. The results of this research revealed that acetylation modification is a crucial factor in regulating the hepatic energy metabolism and lipid synthesis of laying hens. Nevertheless, the impact of lysine acetylation modification on hepatic lipid metabolism in laying hens with FLHS has been rarely reported. Therefore, there is a critical need for further research aimed at elucidating the functional consequences of lysine acetylation in this context. 

In order to explore the potential involvement of lysine acetylation modifications in the liver metabolism of laying hens with FLHS, we conducted an analysis of the hepatic proteome and acetylated proteome in both normal and FLHS-affected birds. In this study, a high-energy and low-protein diet was used to simulate production practices, and the model of FLHS in laying hens was successfully constructed. A comprehensive analysis was conducted to elucidate the hepatic biogenesis in laying hens with FLHS. By comparing the hepatic proteome and acetylated proteome between normal and FLHS-affected birds, better understandings of the function of lysine acetylation in hepatic metabolism and lipid synthesis can be gained in this context. Overall, this research has important implications for improving our understanding of the pathophysiology of FLHS and developing more effective strategies for its prevention and treatment.

## 2. Results

### 2.1. Identification of Hepatic Proteins with Differential Expression between Normal and FLHS

Firstly, the morphology of the liver showed significant hemorrhage and lipid deposition in the livers of the modeled hens (Appendix A). In addition, the oil red O and H&E staining showed lipid deposition and vacuolation in the livers of the modeled hens (Appendix A). These findings confirm the successful construction of FLHS in the model group (MOD). The soluble proteins from livers obtained from the control group (CON) and MOD hens were analyzed in three replicates using quantitative proteomics (Appendix A). The peptides identified by mass spectrometry were found to meet the quality control requirements, with the majority of them ranging from 7 to 20 amino acids in length (Appendix A). In this experiment, 4858 proteins were identified, and a quantification analysis was performed on 3905 of them (Figure 1A). Among the quantified proteins, 364 SDEPs were identified. Of these, 198 were upregulated (*p* ≤ 0.05 and 1.5 ≤ MOD/CON ratio) and 166 were downregulated (*p* ≤ 0.05 and MOD/CON ratio ≤ 0.67) (Figure 1B).

### 2.2. Functional Annotation of SDEPs

The Gene Ontology (GO) annotation analysis (Appendix A) was conducted to explore the biological function of significantly differently expressed proteins (SDEPs). The results of the molecular function revealed that a majority of the downregulated proteins were highly enriched in the peptidase inhibitor and regulator activity (*p* < 0.001), endopeptidase inhibitor and regulator activity (*p* < 0.001), and protein homodimerization activity (*p* < 0.002). Meanwhile, some of the proteins exhibiting upregulation were involved in the acyl-CoA desaturase activity (*p* = 0.01), and pyruvate transmembrane transporter activity (*p* = 0.001; Figure 2A).

The results of the biological process showed that many of the downregulated proteins were significantly enriched in the response to the fatty acid (*p* < 0.001) regulation of cholesterol storage (*p* < 0.001), and cellular amino acid metabolic processes (*p* < 0.001). Meanwhile, the majority of upregulated proteins were involved in the glycerolipid metabolic process (*p* = 0.0075), regulation of lipid storage (*p* < 0.0075), negative regulation of lipid storage (*p* < 0.001), and phospholipid metabolic process (*p* < 0.01; Figure 2B).

The results of the cellular component indicated that the downregulated proteins were significantly enriched in the extracellular space (*p* < 0.001) and extracellular region (*p* < 0.001), while many upregulated proteins were related to the exocyst (*p* < 0.01) and Golgi stack (*p* = 0.04; Figure 2C). Interestingly, some of the downregulated proteins were significantly enriched in the chylomicron (*p* < 0.015) and low-density lipoprotein particle (*p* < 0.015; Figure 2C).

The results of the Kyoto Encyclopedia of Genes and Genomes (KEGG) pathway (Appendix A) revealed that the upregulated proteins were involved in lipid metabolism pathways such as fat digestion and absorption (*p* = 0.02), biosynthesis of the unsaturated fatty acids (*p* < 0.05), and glycerophospholipid metabolism (*p* = 0.01, Figure 2D). Upregulated proteins including acyl-CoA wax alcohol acyltransferase 1 (AWAT1), phospholipid phosphatase 1 (PLPP1), and apolipoprotein A-IV (APOAIV) are associated with fat digestion and absorption, while fatty acid desaturase 1 (FADS1), fatty acid desaturase 2 (FADS2), and the elongation of very long chain fatty acids protein 6 (ELOVL6) are involved in the biosynthesis of unsaturated fatty acids. Phosphate cytidylyltransferase 1 alpha subunit (PCYT1A), PLPP1, and phosphatidylserine decarboxylase 1 (PTDSS1) are involved in glycerophospholipid metabolism. These findings suggested that an abnormal lipid metabolism occurred in the hens with FLHS. Many of the downregulated proteins were associated with the pathways of bile secretion, complement and coagulation cascades, and amino acid metabolism. The metabolism of histidine, tyrosine, glycine, serine, and threonine was enriched with a large number of downregulated proteins. Within the complement and coagulation pathway, fibrinogen alpha chain (FGA), fibrinogen beta chain (FGB), and fibrinogen gamma chain (FGG), which are fibronectin and play an important role as coagulation factors, were significantly downregulated. Additionally, two important transporter proteins, apolipoprotein B (APOB) and cluster of differentiation 36 (CD36), were found to be downregulated and upregulated, respectively.

Furthermore, the results of subcellular localization exhibited many SDEPs that were localized in the cytoplasm and nucleus, with the extracellular space and mitochondria following as the subsequent locations (Figure 2E). In conclusion, the MOD birds exhibited abnormal protein and lipid metabolism in the liver compared to the CON birds.

### 2.3. Differentially Expressed Lysine-Acetylated Proteins in Livers between Normal and FLHS Hens

Similarly, peptide fragments were tested to ensure consistency with the quality control requirements, with the majority of them ranging from 7 to 20 amino acids in length (Appendix A). There were 5166 sites acetylated across 2106 proteins in total. Out of these acetylated sites, quantitation was performed on 2400 sites from 1027 proteins (Figure 3A). Compared to normal laying hens, the liver of FLHS-affected hens showed the identification of 819 proteins with significantly upregulated acetylation levels (PwSUALs) on 1503 lysine sites (*p* ≤ 0.05 and 1.5 ≤ MOD/CON ratio), and 72 proteins with significant downregulated acetylation levels (PwSDALs) on 83 lysine sites (*p* ≤ 0.05 and MOD/CON ratio ≤ 0.67) (Figure 3B). Additionally, the subcellular localization results showed that these proteins were primarily localized in the cytoplasm, followed by the mitochondria and the nucleus (Figure 3C). Most of the PwSUALs were localized in the cytoplasm (Figure 3D), while the PwSDALs were localized in mitochondria (Figure 3E).

### 2.4. Functional Annotation of Lysine-Acetylated SDEPs

The GO annotation analysis (Appendix A) was conducted to explore the biological function of lysine-acetylated proteins. The results of the molecular function indicated that most of the PwSUALs were significantly involved in NAD binding (*p* < 0.01), the lipid transporter activity (*p* < 0.01), and fatty acid binding (*p* < 0.03). PwSDALs were significantly related to binding functions, such as amide binding (*p* < 0.005), coenzyme binding (*p* < 0.015), and cofactor binding (*p* < 0.02); enzyme activity, such as hydro-lyase activity (*p* < 0.005), carbon–oxygen lyase activity (*p* < 0.01), and transferase activity (*p* < 0.005; Figure 4A).

The results of the biological process indicated that many PwSUALs were related to the lipid-related metabolism, including cellular lipid catabolic process (*p* < 0.005), lipid oxidation (*p* < 0.005), neutral lipid metabolic process (*p* < 0.003), aerobic respiration (*p* < 0.003), and cellular respiration (*p* < 0.003). PwSDALs were significantly enriched in lipid oxidation (*p* < 0.004), the cellular lipid catabolic process (*p* < 0.004), and lipid modification (*p* < 0.004); mitochondrial function, such as mitochondrial transmembrane transport (*p* < 0.004), aerobic respiration (*p* < 0.004), and cellular respiration (*p* < 0.004); carboxylic acid metabolism, such as the monocarboxylic acid metabolic process (*p* < 0.004), dicarboxylic acid metabolic process (*p* < 0.004), and tricarboxylic acid metabolic process (*p* < 0.004; Figure 4B).

The results of the cellular component revealed that some of the PwSUALs were significantly related to the ribosome and endoplasmic reticulum, such as cytosolic ribosome (*p* < 0.005), ribosomal subunit (*p* < 0.005), ribosome (*p* < 0.005), and rough endoplasmic reticulum (*p* < 0.005). PwSDALs were significantly enriched in the mitochondrion, such as the mitochondrial matrix (*p* < 0.01), mitochondrial inner membrane (*p* < 0.01), and mitochondrion (*p* < 0.01; Figure 4C).

Furthermore, the analysis of the KEGG pathways (Appendix A) revealed that many PwSUALs were significantly involved in the pathways of the ribosome (*p* < 0.001; Figure 4D), including most members of the RPL and RPS family of ribosomal proteins; TCA cycle (*p* < 0.001), such as aconitase 1 (ACO1), isocitrate dehydrogenase 2 (IDH2), and malate dehydrogenase 1 (MDH1); fatty acid degradation (*p* < 0.01), including acyl-CoA synthetase bubblegum family member 2 (ACSBG2), alcohol dehydrogenase 1 (ADH1), and acyl-CoA synthetase long-chain family member 1 (ACSL1); and the PPAR signaling pathway (*p* < 0.001), such as some members of the fatty-acid-binding protein family. PwSDALs were associated with the pathways of valine, leucine, and isoleucine degradation (*p* < 0.001), such as methylmalonyl-CoA epimerase (MCEE) and acyl-CoA dehydrogenase family member 8 (ACAD8); and fatty acid degradation (*p* < 0.01), such as hydroxyacyl-CoA dehydrogenase subunit alpha (HADHA) and alcohol dehydrogenase 2 (ADH2).

### 2.5. Preferences in Sequence of Lysine Sites

The Motif-x program was employed to analyze the sequence context and preference of acetylated lysine in the protein. The motif results showed different abundances of acetylated lysine sites, with the KacH (15.7%, 716/4572), KacS (14.2%, 650/4572), KacT (11.9%, 544/4572), and KacN (11.8%, 540/4572) motifs being the usual ones (Figure 5A,B). At the +1 position near the Kac site, histidine (H), asparagine (N), serine (S), and threonine (T) residues were highly enriched, while glycine (G), asparagine (N), and threonine (T) were commonly observed at the −1 position. Alanine (F), histidine (H), isoleucine (I), valine (V), and tyrosine (Y) were observed at the −2 position (Figure 5C).

### 2.6. Cluster Analysis Based on Protein Acetylation Levels

Based on the differential acetylation multiples, the acetylated proteins were classified into four categories (Q1–Q4) to explore the relationship between the protein function and the levels of acetylation modification (Figure 6A). A GO classification, KEGG pathway analysis, and clustering analysis were then performed for each group separately to identify correlations between the protein functions and differentially expressed proteins in the comparison groups.

Regarding biological processes (Figure 6B), the PwSDALs in the Q1 category were mainly involved in triglyceride metabolism, mitochondrial gene expression, protein targeting to the endoplasmic reticulum, covalent chromatin modification, and histone modification. The PwSDALs in the Q2 category were enriched in cellular metabolic processes, including cellular catabolism, drug metabolism, and aerobic respiration, as well as lipid metabolism, including lipid modification, fatty acid oxidation, and lipid oxidation. The PwSUALs in the Q3 were related to biological regulation and the regulation of biological processes. Interestingly, no significant enrichment of PwSUALs was observed in the Q4 category.

Concerning cellular components (Figure 6C), the PwSDALs in the Q1 were significantly involved in ribosomes, including the large ribosomal subunit, cytosolic ribosome, and ribosomal subunit, as well as the endoplasmic reticulum, such as the endoplasmic reticulum membrane and subcompartment. The PwSDALs in the Q2 category were significantly enriched in oxidoreductase complexes, oxoglutarate dehydrogenase complexes, mitochondrial cristae, and mitochondrial oxoglutarate dehydrogenase complexes. The PwSUALs in the Q3 were associated with the membrane-enclosed lumen, intracellular organelle lumen, and organelle lumen, while in the Q4 category, the PwSUALs were enriched in the mitochondrial membrane and mitochondrial envelope.

For the molecular function (Figure 6D), the PwSDALs in the Q1 category were enriched in lipid metabolism, including fatty acid binding, lipid transporter activity, and long-chain fatty-acid-CoA ligase activity. The PwSDALs in the Q2 category were associated with fatty acid oxidation-related biological functions, such as 3-hydroxy acyl-CoA dehydrogenase activity, long-chain-enoyl-CoA hydratase activity, and mitochondrial function such as succinate dehydrogenase activity, electron transfer activity, and DNA binding. In the Q3 category, the PwSUALs were correlated with sulfurtransferase activity and thiosulfate sulfurtransferase activity. In the Q4 category, the PwSUALs were related to medium-chain-acyl-CoA dehydrogenase activity and isomerase activity.

For the KEGG pathway (Figure 6E), the PwSDALs in the Q1 category were involved in the ribosome pathway, including ribosomal protein family (RPL and RPS), and the PPAR signaling pathway, including fatty acid binding protein 1 (FABP1), carnitine palmitoyltransferase 1A (CPT1A), and ACSL1. The PwSDALs in the Q2 category were enriched in fatty acid degradation, glycolysis/gluconeogenesis, and some amino acid metabolism, such as glycine, serine, and threonine metabolism, arginine and proline metabolism, and beta-alanine metabolism. The PwSUALs in the Q3 category were related to the sulfur relay system (*p* < 0.001) and sulfur metabolism (*p* < 0.05), including 3-mercaptopyruvate sulfurtransferase (MPST) and thiosulfate sulfurtransferase (TST). The PwSUALs in the Q4 category were related to glutathione metabolism (*p* < 0.05), such as leucine aminopeptidase 3 (LAP3) and glutathione s-transferase mu 2 (GSTM2).

In summary, abnormally elevated acetylation modifications affect the activity of many proteins related to the fatty acid degradation, TCA cycle, ribosome function, and fatty acid oxidation, thereby exacerbating the hepatic lipid accumulation and metabolic disorders in hens with FLHS.

### 2.7. The Network of Acetylated Protein Interactions

Proteins with significantly different levels of acetylation were employed for the construction of a protein–protein interactions (PPI) network to explore their interactions. The network displayed 60 acetylated proteins and three clusters. Proteins enriched in cluster 1 were involved in ribosomal functions, with RPS27A, a key enzyme involved in ribosome structure formation, transcriptional regulation, DNA binding, and protein synthesis, showing the highest degree of connectivity. Moreover, cluster 1 contained fifteen high-interacting PwSUALs, such as ribosomal protein L27a (RPL27A) and ribosomal protein S3a (RPS3A) (Figure 7 and Table 1). Proteins enriched in cluster 2 were associated with the TCA cycle, including seven PwSUALs, such as ACO1, IDH2, and pyruvate dehydrogenase E1 component subunit alpha 2 (PDHA2), and a PwSDAL fumarate hydratase (FH), which displayed high interaction degrees. In cluster 3, four PwSDALs, including acyl-CoA oxidase 1 (ACOX1), acyl-CoA oxidase 2 (ACOX2), enoyl-CoA delta isomerase 2 (ECI2), and enoyl-CoA hydratase and 3-hydroxyacyl CoA dehydrogenase (EHHADH), and three PwSUALs, including acyl-CoA dehydrogenase long chain (ACADL), acyl-CoA synthetase long chain family member 5 (ACSL5), and ACSBG2, were involved in fatty acid degradation.

### 2.8. Comprehensive Analysis of Proteome and Acetyl-Proteome

The change in the liver metabolism from normal to FLHS stage in hens was investigated through a comprehensive analysis of proteomic and acetyl-proteomic data. After screening the data, the proteins detected simultaneously were presented in a tabular format (Table 2). The findings indicated that many proteins subjected to acetylation modifications were downregulated. These proteins were mainly involved in amino acid metabolism, unsaturated fatty acid biosynthesis, and the PPAR signaling pathway. Proteins involved in the PPAR signaling pathway included fatty acid binding protein 4 (FABP4), CD36, perilipin 2 (PLIN2), and fatty acid binding protein 7 (FABP7), while the proteins associated with the biosynthesis of unsaturated fatty acids were FADS1 and FADS2. The amino acid metabolism pathways included tyrosine, histidine and glycine, serine, and threonine metabolism. The proteins involved in these pathways were urocanate hydratase 1 (UROC1), histidine ammonia-lyase (HAL), 4-hydroxyphenylpyruvate dioxygenase (HPD), and fumarylacetoacetate hydrolase (FAH). These results suggest that altered protein abundance and acetylation levels influence the amino acid and lipid metabolism in FLHS-affected hens.

### 2.9. Acetylation Analysis of Lipid Metabolism in Laying Hens with FLHS

The above results strongly suggest that abnormal lipid metabolism occurred in FLHS-affected laying hens, associated with the upregulation of protein acetylation. Thus, we selected a number of important proteins for a detailed analysis of the acetylation results. The proteins significantly regulated by acetylation were summarized, and it was found that the protein functions affected in the livers of laying hens were mainly involved in fatty acid oxidation and lipid transport (Figure 8 and Figure 9, and Table 3). The main proteins affected in lipid transport were microsomal triglyceride transfer protein (MTTP), fatty acid binding protein (FABP), and APOB proteins. The reduced activity of these proteins following acetylation modification resulted in blocked lipid transport and the abnormal accumulation of hepatic lipids, in line with the findings of others. In contrast, proteins with increased acetylation during fatty acid oxidation were ACSL5, ACSBG2, carnitine palmitoyltransferase 1 (CPT1), and ACOX1. These proteins suffered from acetylation and were unable to regulate the mitochondrial uptake of long-chain fatty acids for beta-oxidation, leading to a reduction in acetyl coenzyme A. This, in turn, resulted in less raw material for the TCA cycle, and reduced the efficiency of the fatty acid metabolism by hepatocytes.

## 3. Discussion

FLHS is a significant issue in the poultry industry, and there is great interest in understanding the changes in the liver lipid metabolism that occur in FLHS-affected hens [10]. However, the detailed mechanisms that regulate hepatic lipid changes from normal to FLHS stages in laying hens are still unclear. To address this, we constructed FLHS-affected hens by simulating a high-energy and low-protein diet commonly used in the production practice. Subsequently, we analyzed the hepatic proteome and acetylated proteome of normal and FLHS laying hens to explore the involvement of acetylation in the progression of FLHS. Compared to normal laying hens, upregulated proteins in FLHS-affected hens were mainly associated with fat digestion and absorption, unsaturated fatty acid biosynthesis, and glycerophospholipid metabolism, while downregulated proteins predominantly participated in bile secretion, complement and coagulation cascades, and amino acid metabolism. Additionally, this research revealed that significant acetylated proteins in FLHS hens had a strong correlation with the ribosome function, TCA cycle, and fatty acid degradation, while significant deacetylated proteins were involved in lipid oxidation and catabolism, as well as fatty acid degradation.

Increasing evidence from numerous studies indicates that acetylation modifications are significant regulators of a broad spectrum of metabolic activities in the liver, achieved through their acetylation of key sites on metabolically relevant proteins affecting enzymatic activity [11,12]. Of primary interest is the hepatic lipid metabolism in hens with FLHS. In this research, the proteins associated with altered levels of the acetylation of lipid metabolism clearly participated in fatty acid degradation and lipid transport. The PwSUALs associated with fatty acid degradation include ACSL5, ACSBG2, ADH1, and ACADL. It has been shown that the deacetylation of ACSL5 promotes fatty acid oxidation and inhibits lipid accumulation [13]. Therefore, the upregulation of acetylation levels of ACSL5 leads to the inhibition of fatty acid oxidation. ACSBG2 facilitates the synthesis of cellular lipids by activating long-chain fatty acids, while also generating acetylated fatty acids through their oxidative degradation [14]. Although ADH1 has been shown to be reduced in the livers of patients with fatty liver, little has been reported on whether it is regulated by acetylation [15]. In contrast, hepatic ACADL acetylation levels were significantly increased in HFD mice fed fructose, resulting in a reduced fat metabolism [5]. The PwSDALs associated with fatty acid degradation include ACOX1, ECI2, and EHHADH, two important enzymes that catalyze fatty acid degradation and produce acyl-coenzyme A. The acetylation of ACOX1 was found to be downregulated in granulosa cells with disturbed lipid metabolism, and the acetylation of EHHADH at various lysine residues was found to increase its enzymatic activity and promote fatty acid oxidation [12,16]. Thus, the downregulation of EHHADH acetylation in chicken liver would reduce fatty acid oxidation. The investigators found that the acetylation level of the fatty acid degradation-related enzyme ECI2 was significantly upregulated and activated for expression following the inhibition of deacetylase activity [17]. These results suggest that fatty acid oxidation is inhibited through the modification of the acetylation and deacetylation of fatty acid degradation-associated proteins in FLHS-affected chicken, affecting their activity or expression. The PwSUALs involved in lipid transport include hepatocyte nuclear factor 4 alpha (HNF4α), MTTP, APOB, and FABP1. As a member of the nuclear receptor superfamily, HNF4α broadly regulates the expression of genes related to the metabolic pathways of glucose, cholesterol, and fatty acids [18,19]. SIRT2 has been shown to prevent hepatic steatosis and metabolic disorders through the deacetylation of HNF4α, and the upregulation of HNF4α acetylation levels increases hepatic lipid accumulation [20]. The investigators found that HFD-induced hepatic FABP1 acetylation levels were downregulated in mice after AEGL treatment, resulting in significant improvement in hepatic steatosis [21]. Thus, acetylation modification inhibits fatty acid degradation and lipid transport through its action on related proteins, and it can be inferred that this is one of the important reasons for the development of hepatic lipid deposition in FLHS laying hens.

In addition to lipid metabolism, vigorous amino acid and energy metabolisms are essential for maintaining egg-laying activities in hens. The normal metabolism of methionine and lysine is particularly important. Acetylation histology results show that the acetylation levels of most proteins responsible for methionine and lysine metabolism are upregulated. Although there are few reports on how acetylation modifications regulate amino acid metabolism in laying hens, it can be hypothesized that acetylation upregulation reduces the amino acid metabolism in hens with FLHS, which leads to reduced egg production. Pyruvate metabolism is a representative process enriched with acetylated upregulated proteins in energy metabolism. Data show that the acetylation levels of pyruvate carboxylase (PC) and phosphoenolpyruvate carboxykinase 2 (PCK2), which limit the rate of gluconeogenesis, are noticeably elevated in the liver of fatty liver hens [22]. Pyruvate dehydrogenase e1 component subunit alpha (PDHA1) was found to be hyperacetylated in the study. The deacetylation of PDHA1 enhanced the activity of PDH [23]. This suggests that the catalytic activity of PDH is inhibited through the acetylation of PDHA1, resulting in an abnormal energy metabolism in hens with FLHS. Furthermore, two important lactate dehydrogenases, lactate dehydrogenase A (LDHA) and lactate dehydrogenase B (LDHB), suffered deacetylation and acetylation, respectively, in fatty liver laying hens. LDHA is a key protein in the lactate synthesis pathway, and its acetylation level has been negatively correlated with the expression level and reduced LDHA enzyme activity and protein levels [24,25]. Therefore, elevated LDHA activity in fatty-livered hens is observed. LDHB is the key protein that facilitates the conversion of lactate to pyruvate and NADH. In human and mouse non-alcoholic simple fatty livers as well as non-alcoholic steatohepatitis livers, hyperacetylation of LDHB has been linked to lactate accumulation [26,27]. The impairment of lactate clearance due to the PCAF-dependent acetylation of LDHB was responsible for hepatic lipid accumulation and inflammatory responses [28]. Thus, LDHA and LDHB may be potential therapeutic targets for FLHS.

In our study, we found more proteins regulated by acetylation in the TCA cycle. ACO1, IDH2, MDH1, and PDHA2 are related to the conversion of NADP+ to NADPH through the TCA cycle [29,30]. However, the upregulation of acetylation has been shown to reduce their catalytic activity. Oxidative phosphorylation is the main source of energy for aerobic cellular activities and the main pathway for ATP production [31]. The acetylation of oxidative-phosphorylation-related proteins inhibits the activity of electron respiratory chain complexes, thereby blocking the conversion of NADPH to ATP. Similarly, we observed an increase in the acetylation levels of proteins involved in oxidative phosphorylation, such as ATP synthase peripheral stalk subunit H (ATP5H), succinate dehydrogenase flavoprotein subunit A (SDHA), and succinate dehydrogenase iron–sulfur subunit B (SDHB). It has been shown that the acetylation of SDHA inhibits the activity of the SDH complex, leading to intracellular succinate accumulation [32]. These results suggest that acetylation modifications inhibit the TCA cycle and oxidative phosphorylation, impairing the energy metabolism in hens with FLHS. 

Acetylated proteins are vital for regulating hepatic lipid metabolism through interactions [33,34]. Interestingly, our study identified three clusters in the PPI network, including fatty acid degradation, ribosome function, and the TCA cycle. Although ribosomes are not directly involved in lipid metabolism, they synthesize numerous essential proteins involved in this process [35]. Furthermore, the acetylation of ribosomal proteins increases the binding efficiency of ribosomes to mRNA, leading to an increased frequency of translation pauses in ribosomes when excessive binding occurs [36,37]. Additionally, it was found that the acetylation of ribosomal proteins decreases the binding capacity of rRNA [38,39]. The lysine contents of ribosomal and non-ribosomal proteins were also analyzed, and it was significantly higher in the former, suggesting that acetylation may affect the assembly and translation efficiency of ribosomes. Among the proteins in the network, acetylated ACOX1, ACOX2, ECI2, and EHHADH are integral enzymes for the regulation of fatty acid degradation. The PPI network also contains numerous proteins that are associated with the TCA cycle, and these proteins contribute to the final conversion of acetyl coenzyme A, a substrate produced by fatty acid oxidation, into ATP. Therefore, blockage of the TCA cycle is also a crucial factor leading to decreased fatty acid oxidation.

As a result of the cluster analysis of differential acetylated proteins, histone modification and covalent chromatin modification and DNA binding are both indicated as one of the biological processes and molecular functions, respectively. Changes in histone modification and covalent chromatin modification can be interpreted as these proteins being involved in the regulation of the chromatin state. Histone modification refers to the chemical modifications, such as acetylation and methylation, on histone molecules, which can alter the compactness of chromatin and thereby affect gene expression [40,41]. Covalent chromatin modification refers to modifications on non-coding DNA regions that interact with histone molecules, such as methylation and phosphorylation, which can affect the chromatin structure and stability, thereby impacting gene expression [42]. Therefore, changes in histone modification and covalent chromatin modification may indicate involvement in the regulation of the chromatin state and may lead to changes in the expression of certain genes. These clustering results of differentially acetylated proteins in FLHS-affected hens provide a preliminary clue to help researchers explore how these proteins are involved in the regulation of the chromatin state and understand their impact on gene expression. Generally, the acetylation modification of a protein increases its affinity for DNA, promoting its binding to DNA [43]. This is because acetylation modification can alter the protein’s charge state, making it more likely to interact with the negative charges in DNA [44]. In our research, RPS27A showed the highest degree of connectivity in the PPI network. Studies have shown that RPS27A can bind to specific DNA sequences, and affect DNA replication and repair processes [45]. In addition, acetylation modification can also affect the structure and function of chromatin [46,47]. By modifying histones with acetylation, the compactness of chromatin can be altered, thereby affecting the accessibility and readability of DNA. Overall, the acetylation modification of proteins increases their affinity for DNA, may enhance their function in chromatin, and may affect the accessibility and readability of DNA in hens with FLHS.

After analyzing the proteomics and acetyl-proteomics data, we found that the number of proteins detected simultaneously was not large, suggesting that acetylation modifications predominantly affect the protein activity rather than expression, consistent with its role as a post-translational modification. Notably, some proteins, such as HADHA, alcohol dehydrogenase 4 (ADH4), and carnitine palmitoyltransferase 2 (CPT2) in fatty acid degradation, and IDH2, FH, and aconitase 2 (ACO2) in the TCA cycle, can undergo acetylation and deacetylation concurrently at distinct lysine locations, resulting in the complex regulation of their biological activities. This highlights the intricate mechanisms underlying the acetylation of proteins related to hepatic lipid and energy metabolism.

## 4. Materials and Methods

### 4.1. Animals and Sampling

A total of 120 sixty-one-week-old Jingfen 6 hens were randomly selected and divided into two groups: CON and MOD. A one-way completely randomized design was used, with twelve replicates and five hens in each replicate. The vaccination and immunization procedures were carried out according to standard procedures. One chicken was randomly selected from each of the two replicates, and six chickens from each group, for a total of twelve chickens to be sampled. The sampled chickens were dissected in a sterile environment, and their weights were recorded. The abdominal cavity was then dissected, and the liver was removed intact and photographed for preservation. After saline rinsing, the surface of the liver was dried with filter paper, weighed, and recorded. For each laying hen, a portion of the liver tissue of the same size was excised at the edge of the liver lobules using a scalpel. This portion was fixed in 4% paraformaldehyde solution, and pathological sections were prepared in the laboratory for further oil red O and H&E staining.

### 4.2. Protein Extraction and Trypsin Digestion 

The liver samples were chilled in liquid nitrogen and ground into powder before being transferred to centrifuge tubes. The tubes were treated with an appropriate amount of lysis buffer, which included 8 M urea and 1% protease inhibitor cocktail. For acetyl-proteome experiments, 3 μM Tris-saline-azide (TSA) and 50 mM N-acetylmuramic acid (NAM) were also included in the lysis buffer. The samples were subjected to three rounds of treatment using a high intensity ultrasound processor (Scientz) while kept on ice. Following centrifugation at 12,000× *g* for ten minutes at 4 °C, the supernatant was gathered. Subsequently, the protein concentration was determined using the BCA kit (Applygen, Beijing, China).

Next, the extracted proteins needed to be digested. The protein samples underwent reduction with 5 mM dithiothreitol at 56 °C for 30 min, followed by alkylation with 11 mM iodoacetamide in the dark at 25 °C for 15 min. Subsequently, the samples were washed using 100 mM of TEAB buffer to decrease the urea concentration to below 2 M. After adding trypsin to the samples at a 1:50 trypsin/protein mass ratio, the first digestion was conducted overnight. Subsequently, the second digestion was conducted for 4 h at a 1:100 ratio. The resulting peptides were purified using a C18 solid phase extraction (SPE) column (Phenomenex, Torrance, CA, USA) [48].

### 4.3. TMT Labeling and HPLC Fractionation

The trypsinized peptides were solubilized in 0.5 M TEAB. They were subsequently labeled with the appropriate TMT reagents (Thermo Fisher Scientific, Waltham, MA, USA) and incubated at 25 °C for 2 h. After desalting 5 microliters of the labeled samples for an MS analysis to assess the labeling efficiency, 5% hydroxylamine was put into quenching the reaction. Subsequently, the treated samples were purified using a Strata X C18 SPE column (Phenomenex, Torrance, CA, USA) and dried using a vacuum centrifuge [49].

The samples were fractionated using high-performance liquid chromatography (HPLC) on an Agilent 300 Extend C18 column. Briefly, the peptides were eluted using a 2–60% acetonitrile gradient in 10 mM ammonium bicarbonate over 80 min, resulting in 80 fractions. These fractions were consolidated into nine fractions and freeze-dried using vacuum centrifugation for analysis by LC-MS/MS.

### 4.4. Acetylation Modification Enrichment Assay

To enrich the acetylated peptides, they were dissolved in NETN buffer and subsequently added to pre-washed anti-acetylation resin (Jingjie Biotechnology Co., Hangzhou, China). The mixture was kept with gentle shaking at 4 °C overnight for incubation. After that, the resin was subjected to four washes using an NETN buffer and two additional washes with deionized water. The resin was then subjected to elution using 0.1% trifluoroacetic acid, and the resulting eluate was vacuum freeze-dried. Finally, desalting of the peptides was carried out with ZipTips C18 (Sigma, Shanghai, China) in preparation for subsequent LC-MS/MS analysis.

### 4.5. LC-MS/MS Analysis and 3D Mass Spectrometer

The enriched peptides were dissolved in solvent A (0.1% formic acid and a mixture of 2% acetonitrile and water) and introduced to a reverse phase column for loading, followed by gradient separation with solvent B on an EASY-nLC 1000 system (Thermo Fisher Scientific, Waltham, MA, USA). The gradient used the following settings: 0–60 min at 5–25% B, 60–82 min at 25–35% B, 82–86 min at 35–80% B, and 86–90 min at 80% B. The flow rate of the mobile phase remained constant at 450 nL/min throughout the process.

The isolated peptides were then introduced into a nanoelectrospray ion source for analysis on a Q ExactiveTM HF-X (Thermo Fisher Scientific, Waltham, MA, USA) with an electrospray voltage of 2.0 kV. Intact peptides were full-scanned with a resolution of 60,000 in the range of 350–1600 *m*/*z*. Intact peptides were full-scanned at a full resolution of 60,000 within the 350–1600 *m*/*z* range. Fragments were generated by HCD fragmentation using 28% normalized collision energy (NCE), and then detected by Orbitrap at a resolution of 30,000. LC-MS/MS analysis was performed on the top 20 most abundant precursors, with a dynamic exclusion time of 30 s applied to prevent re-selection. AGC was employed with a target value of 1E5. Additionally, the fixed first mass of 100 *m*/*z* was set for the analysis [50].

### 4.6. Database Search 

The mass spectrometry data were analyzed using the MaxQuant search engine (version 1.6.15.0) [51]. Searches were performed in the Gallus_gallus_uniprot_9031 database, with the addition of a reverse decoy database and a common contamination library to calculate the false positive rate (FPR) due to random matching and to eliminate the effects of contaminating proteins. The enzymatic cleavage mode was set to trypsin/P with a maximum of 2 missed cleavage sites allowed. The First search and Main search were configured to use a precursor ion mass error of 20 ppm and 5 ppm. Additionally, the mass error for fragment ions was established at 0.02 Da. Fixed modification was established as cysteine alkylation, while variable modifications encompassed methionine oxidation, protein N-terminal acetylation, deamidation, and lysine acetylation. Finally, the FPR was adjusted to 1%.

### 4.7. Motif Analysis

The motif features of the modified sites were investigated by the Motif-x software (version 5.0.2) [52]. A 21-mer sequence model was constructed to analyze the modification sites, including 10 upstream and 10 downstream sites of all identified sites. The modified peptides were considered to have a motif when the count of peptides in a specific characteristic sequence exceeded 20 and the *p*-value ≤ 0.01. The NetSurfP-2.0 software (http://www.cbs.dtu.dk/services/NetSurfP-2.0/, accessed on 10 September 2022) was uses to examine the positioning of lysine residues that were either acetylated or non-acetylated within the protein secondary structures.

### 4.8. Annotation Methods and Functional Enrichment

As previously described, the GO annotation of proteins with the UniProt-GO database were divided into three distinct groups: biological processes, cellular compartments, and molecular functions [53]. To annotate the protein pathways, the KEGG database and the KEGG online service tool KAAS (version 2.0) were used to perform the initial annotation, and then they employed the KEGG mapper (www.kegg.jp/kegg/mapper.html, accessed on 10 September 2022). KAAS was another KEGG online service tool to map the annotation results to the KEGG route database. This approach helped to avoid redundancy in the pathway analysis. For the GO and KEGG results, a two-tailed Fisher’s exact test was used to assess the extent of enrichment of differentially expressed proteins across all identified proteins. Significance was determined by a corrected *p*-value ≤ 0.05 for both GO and KEGG. Finally, the subcellular localization of proteins was determined by the subcellular localization prediction software Wolfpsor (version 0.2).

To enhance the hierarchical clustering analysis that is based on the functional classification of differentially acetylated proteins, we categorized them into four categories (Q1–Q4) according to their differential expression ploidy: Q1 (MOD/CON ratio > 2.0), Q2 (1.5 < MOD/CON ratio ≤ 2.0), Q3 (0.5 < MOD/CON ratio ≤ 0.667), and Q4 (0 < MOD/CON ratio ≤ 0.5). The cluster members were then visualized by a heatmap using the heatmap.2 function in the R package plots (v.2.0.3).

### 4.9. Protein–Protein Interactions Network Analysis

To identify the PPIs, all accession numbers or sequences of the differentially expressed proteins were searched against version 11.0 of the STRING database. Interactions were specifically chosen from within the dataset’s proteins, with any external candidates being excluded. Interactions were included based on their confidence score as defined by the STRING database. Specifically, interactions with a confidence score of ≥0.7 were considered high confidence. The “networkD3” package in R was utilized to visualize the interaction network obtained from STRING.

### 4.10. Statistical Analysis

The statistical significance of differences in the protein expression between the MOD and CON groups were evaluated using a *t*-test (two-sample, two-tailed). The mean quantitative values of the same protein in both MOD and CON were compared by calculating their ratio. Differentially expressed proteins with a MOD/CON ratio ≤ 1/1.5 or MOD/CON ratio ≥ 1.3 and a *p*-value ≤ 0.05 were deemed SDEPs. For the analysis of differences in the acetylation modification levels, the same statistical analysis was applied. All statistical analyses were conducted by SPSS v23.0 (IBM, Chicago, IL, USA). Differences with a *p*-value ≤0.05 were considered to be statistically significant.

## 5. Conclusions

In conclusion, this research presents the hepatic comprehensive proteomic and acetylated proteomic profile of hens with FLHS. Altered physiological conditions in the liver of FLHS hens lead to upregulated proteins, such as AWAT1, PLPP1, and APOAIV, which enhance the digestion and absorption of fat in hens with FLHS. Downregulated proteins, such as APOB, affect the transport function of hepatocytes. Furthermore, acetylated PwSUALs and PwSDALs have regulatory effects on lipid transport and fatty acid oxidation, highlighting the importance of acetylation modifications and providing potential therapeutic targets. Inhibited lipid transport leads to lipid deposition in the liver, while downregulated fatty acid oxidation results in a reduction in acyl-CoA. This reduction leaves the TCA cycle and following oxidative phosphorylation without a sufficient substrate, interferes with the normal energy metabolism in hens with FLHS. Additionally, various acetylated proteins are related to amino acid metabolism, ribosome function, and the TCA cycle. Altogether, these factors affect hepatic lipid metabolism, exacerbating lipid accumulation and metabolic disorders in FLHS-affected layers. This research provides insight into the impacts of regulating hepatic acetylation on hepatic lipid metabolism in laying hens. These findings are crucial in the search for potential therapeutic approaches.

## Figures and Tables

**Figure 1 ijms-24-08491-f001:**
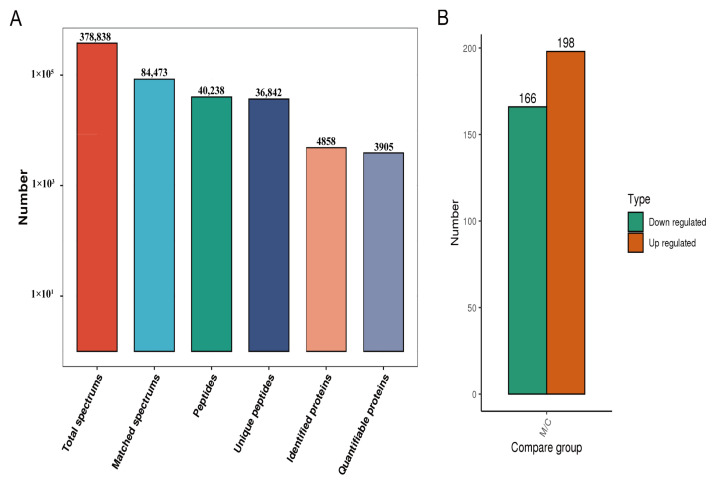
Summary of the hepatic SDEPs in normal and FLHS-affected laying hens. (**A**) Statistical summary of mass spectrometry data. (**B**) Statistics of SDEPs.

**Figure 2 ijms-24-08491-f002:**
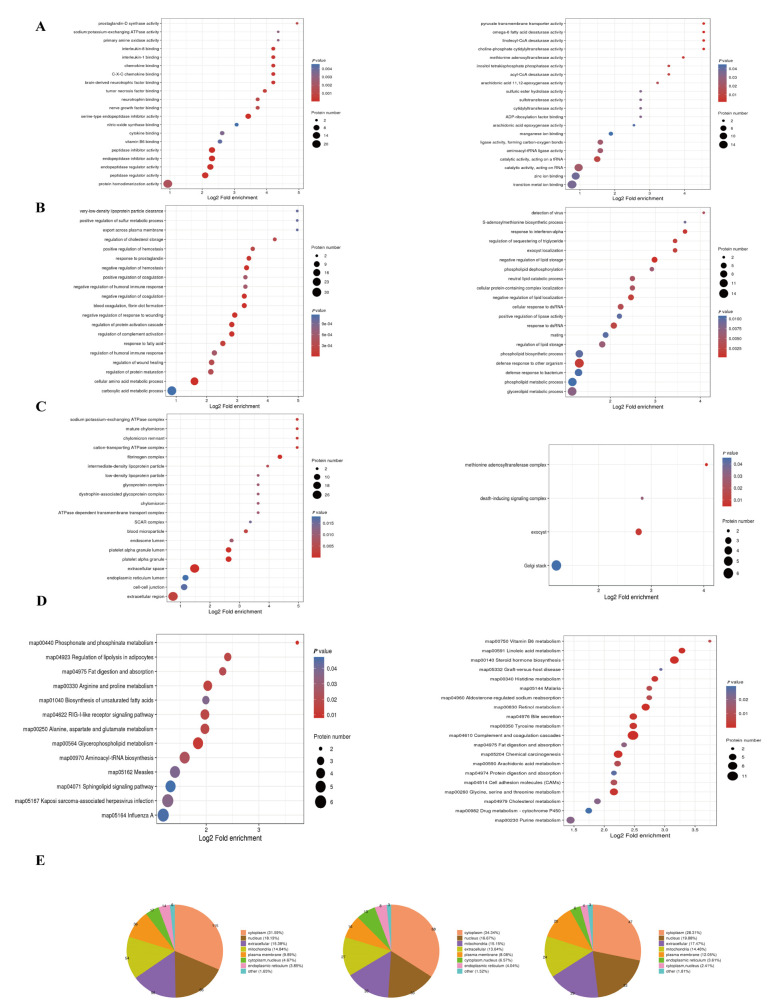
The GO and KEGG pathway enrichment analyses (left box: downregulated proteins; right box: upregulated proteins), and subcellular localization of SDEPs. (**A**) Biological process. (**B**) Cellular component. (**C**) Molecular function. (**D**) KEGG pathway enrichment analysis of SDEPs. (**E**) Subcellular localization of SDEPs (from left to right: all, upregulated, downregulated).

**Figure 3 ijms-24-08491-f003:**
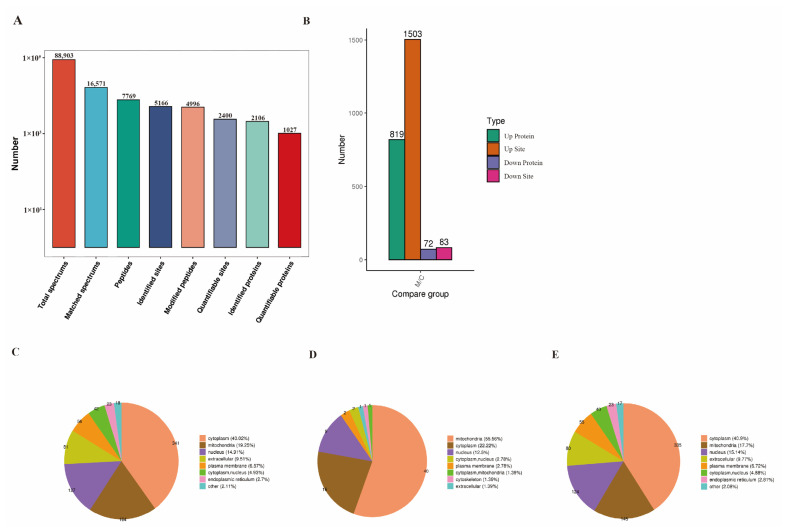
Summary of the hepatic acetylated proteins in normal and FLHS-affected laying hens. (**A**) Statistical summary of mass spectrometry data. (**B**) Statistics of acetylated proteins. (**C**) Subcellular localization of all acetylated proteins. (**D**) Subcellular localization of PwSUALs. (**E**) Subcellular localization of PwSDALs.

**Figure 4 ijms-24-08491-f004:**
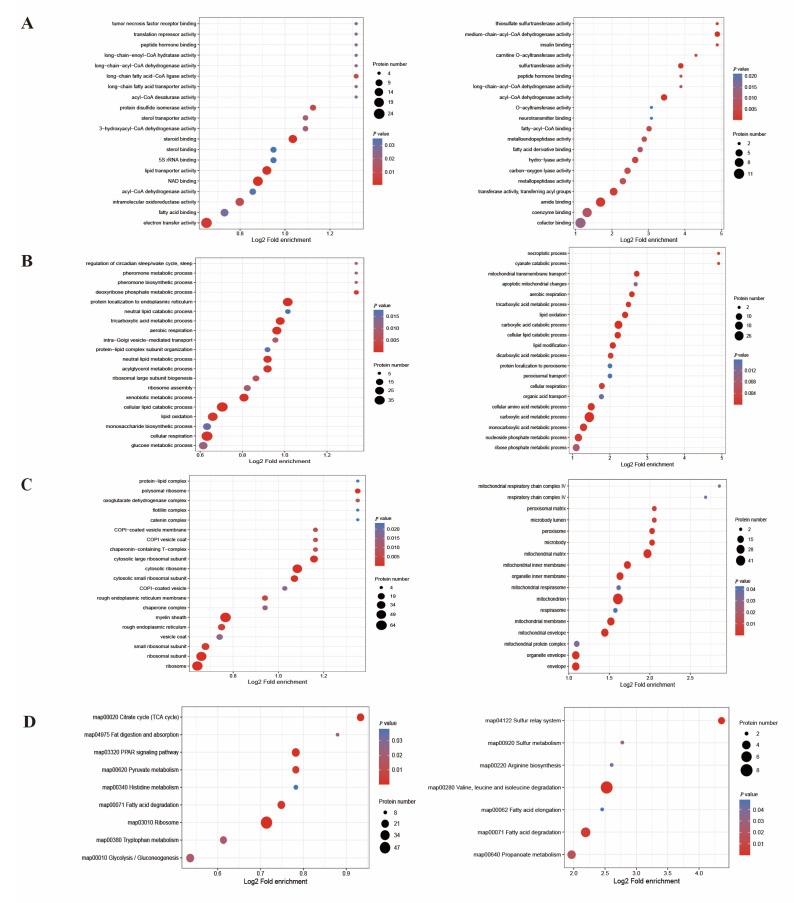
GO and KEGG pathway enrichment analyses of acetylated proteins (left box: upregulated proteins; right box: downregulated proteins). (**A**) Biological process. (**B**) Cellular component. (**C**) Molecular function. (**D**) KEGG pathway.

**Figure 5 ijms-24-08491-f005:**
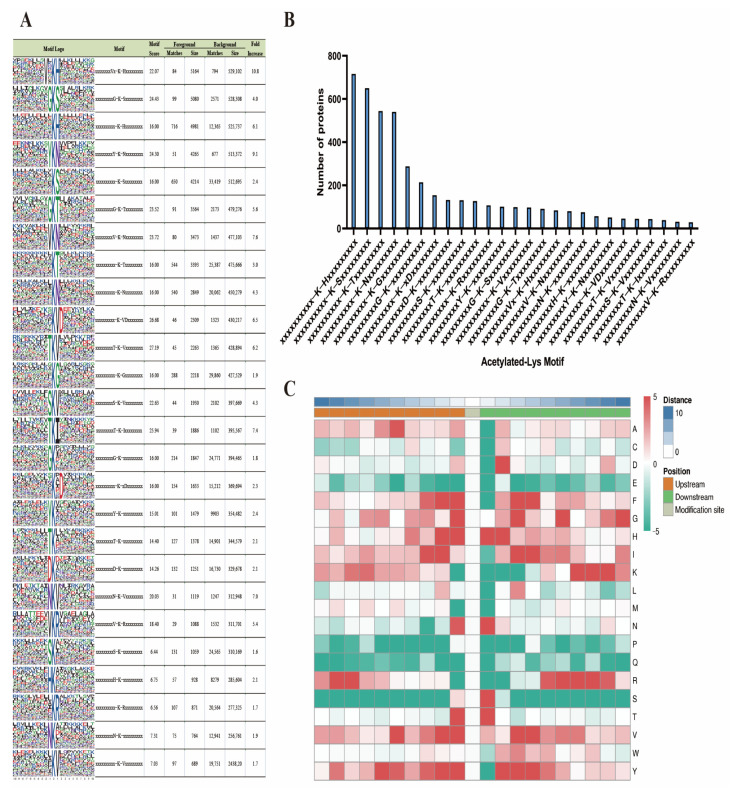
Preference for acetylation site sequences. (**A**) Probable sequence motifs of acetylation sites. (**B**) The count of identified peptides containing acetylated lysines and their probable motifs. (**C**) A heatmap was generated to display the relative frequencies of amino acids (A—Alanine, C—Cysteine, D—Aspartic acid, E—Glutamic acid, F—Phenylalanine, G—Glycine, H—Histidine, I—Isoleucine, K—Lysine, L—Leucine, M—Methionine, N—Asparagine, P—Proline, Q—Glutamine, R—Arginine, S—Serine, T—Threonine, V—Valine, W—Tryptophan, Y—Tyrosine) at specific positions, highlighting the enrichment (red) or depletion (green) of amino acids flanking the acetylated lysine in hepatic proteins.

**Figure 6 ijms-24-08491-f006:**
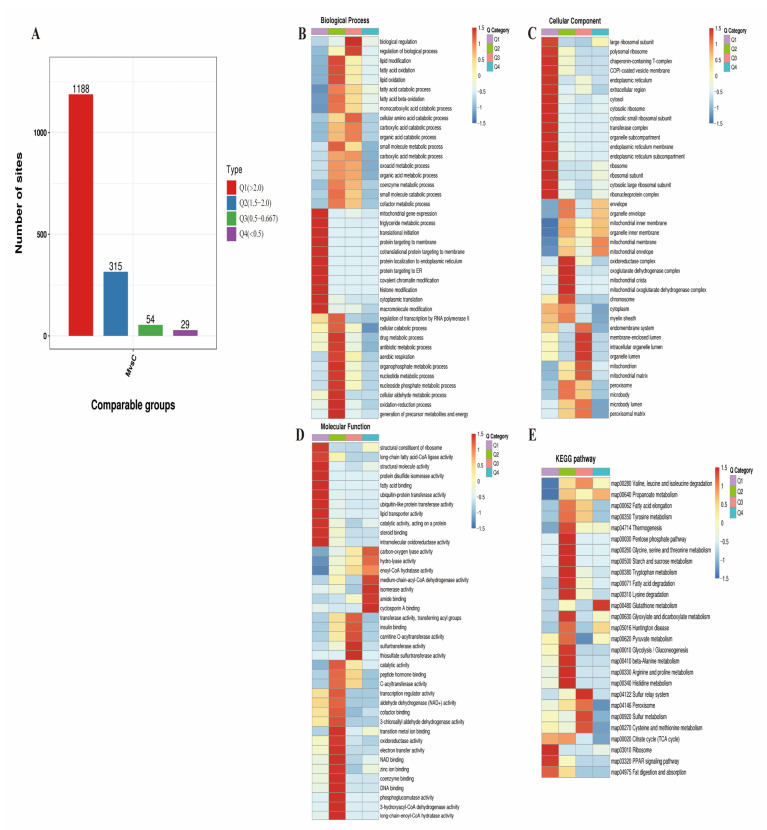
Cluster analysis of differential acetylated proteins based on their differential acetylation multiples (four clusters: Q1, Q2, Q3, Q4). (**A**) Distribution of proteins with differential acetylation levels. (**B**) Biological process. (**C**) Cellular component. (**D**) Molecular function. (**E**) KEGG pathway.

**Figure 7 ijms-24-08491-f007:**
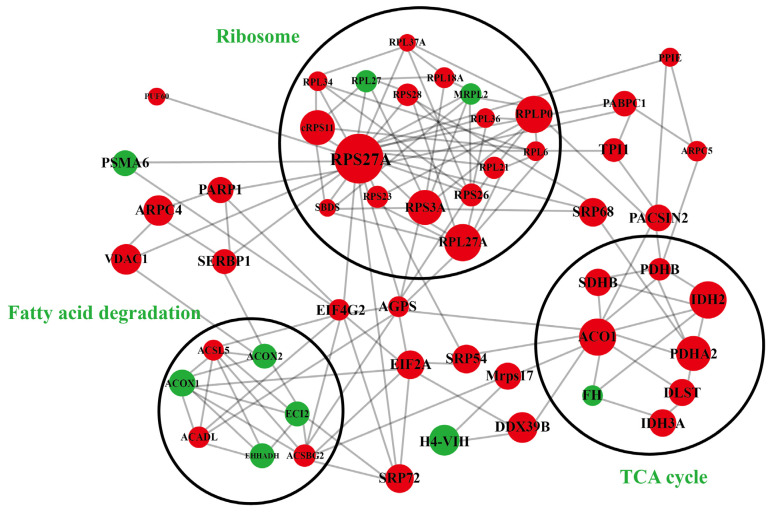
PPI network of differential acetylated proteins.

**Figure 8 ijms-24-08491-f008:**
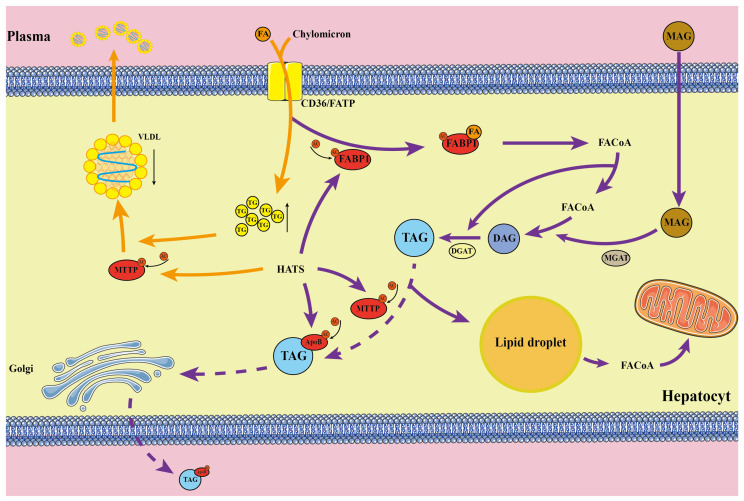
Effects of acetylated proteins on lipid metabolism.

**Figure 9 ijms-24-08491-f009:**
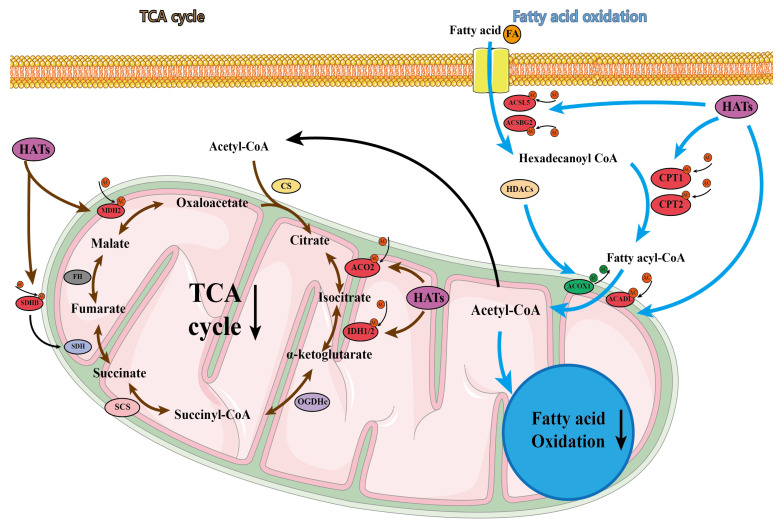
Effects of acetylated proteins on fatty acid oxidation and TCA cycle.

**Table 1 ijms-24-08491-t001:** Top 30 hub-proteins in PPI network.

Protein Accession	Degree	Position	Gene Name	Regulated Type	Ratio
P79781	131	6	RPS27A	Up	2.618
Q90875	85	165	ACO1	Up	3.088
A0A1D5P8I7	84	91	IDH2	Up	1.649
P47826	84	79	RPLP0	Up	3.472
F1NBX4	84	92	RPL27A	Up	1.757
F2Z4K7	83	46	RPS3A	Up	2.299
Q98TH5	83	136	cRPS11	Up	2.959
Q5F426	83	90	PDHA2	Up	1.614
A0A1L1RX65	81	326	IDH3A	Up	1.689
Q9YHT2	81	90	SDHB	Up	3.873
F1NY37	81	653	ACOX1	Down	0.458
E1C5V6	80	671	ACOX2	Down	0.638
P61355	79	27	RPL27	Down	0.057
F1NDC2	79	25	RPS23	Up	2.632
A0A1D5PQV3	79	288	DLST	Up	1.540
A0A1D5PNC5	78	308	ECI2	Down	0.601
A0A3Q3AV17	78	393	EHHADH	Down	0.639
F1NYE5	78	332	SERBP1	Up	2.695
F1NI43	77	645	ACSBG2	Up	3.121
Q5ZM66	77	82	RPS26	Up	1.828
A0A1D5P1U2	77	184	PDHB	Up	3.055
R4GIQ2	77	43	RPL21	Up	3.876
A0A3Q2U825	77	16	RPS28	Up	2.786
F1NNV2	76	633	ACSL5	Up	1.789
F1P5J7	76	150	AGPS	Up	2.049
F1NC38	76	339	ACADL	Up	2.901
F1NPD3	76	12	RPL18A	Up	2.174
F1NY85	76	152	MRPL2	Down	0.398
Q5ZLD1	75	58	FH	Down	0.409
F1NQU7	75	228	EIF4G2	Up	2.160

**Table 2 ijms-24-08491-t002:** The proteins were detected in both proteome and acetyl-proteome.

Protein Accession	Gene Name	Regulated Type (Protein)	Position	Regulated Type (Acetylation)	Pathway
F1NT18	CYP3A5	Down	158	Up	Steroid hormone biosynthesis
E1C9D0	DDO	Up	206	Up	Peroxisome
F1NBE3	GCAT	Down	279	Up	Glycine, serine, and threonine metabolism
F1NTK1	PSAT1	Down	274	Up	Glycine, serine, and threonine metabolism
E1BWX4	AASS	Down	287	Up	Lysine degradation
E1BWX9	UROC1	Down	304	Up	Histidine metabolism
F1NZQ9	HAL	Down	346	Up	Histidine metabolism
F1NTZ0	LOC100857280	Down	335	Up	Tyrosine metabolism
F1NVJ0	HPD	Down	153	Up	Tyrosine metabolism
F1NYW8	FAH	Down	320	Up	Tyrosine metabolism
F1NHQ3	SELENOI	Up	244	Up	Phosphonate and phosphinate metabolism
Q5ZM65	PTDSS1	Up	281	Up	Glycerophospholipid metabolism
O73888	HPGDS	Down	43	Up	Chemical carcinogenesis
A0A1D5NUN3	ACSS2	Up	540	Up	Glyoxylate and dicarboxylate metabolism
A0A3S5ZP86	FADS1	Up	473	Up	Biosynthesis of unsaturated fatty acids
A6NAB8	FADS2	Up	28	Up	Biosynthesis of unsaturated fatty acids
A0A1D5NYF3	FABP4	Up	59	Up	PPAR signaling pathway
F1NER9	CD36	Up	282	Up	PPAR signaling pathway
F7BYG6	PLIN2	Up	109	Up	PPAR signaling pathway
Q05423	FABP7	Down	59	Up	PPAR signaling pathway
A0A1D5PU80	ATP1A1	Down	492	Up	Proximal tubule bicarbonate reclamation
A0A1D5PNC5	ECI2	Up	308	Down	Peroxisome
F1NV02	APOB	Down	2502	Up	Fat digestion and absorption
A0A3Q2U8Y5	TCN2	Up	294	Up	Vitamin digestion and absorption
O73888	HPGDS	Down	43	Up	Chemical carcinogenesis
H9L0D7	WASF2	Down	101	Up	Choline metabolism in cancer

**Table 3 ijms-24-08491-t003:** Significantly acetylated lipid-metabolism-related proteins in hens with FLHS.

Protein Accession	Position	Gene Name	Regulated Type	Ratio	Pathway
Q6B842	749	CPT1	Up	2.38	Fatty acid oxidation
F1P1U3	489	CPT2	Up	1.823	Fatty acid oxidation
F1NNV2	633	ACSL5	Up	2.213	Fatty acid oxidation
F1NI43	645	ACSBG2	Up	3.45	Fatty acid oxidation
F1NC38	339	ACADL	Up	2.901	Fatty acid oxidation
F1NY37	653	ACOX1	Down	0.458	Fatty acid oxidation
E1BVT3	327	MDH2	Up	4.291	TCA cycle
Q9YHT2	90	SDHB	Up	3.873	TCA cycle
F1NPG2	212	IDH1	Up	4.51	TCA cycle
A0A1D5P8I7	393	IDH2	Up	2.034	TCA cycle
Q5ZMW1	467	ACO2	Up	2.814	TCA cycle
A0A1D5PVM0	349	MTTP	Up	9.763	Lipid transport
Q90WA9	121	FABP1	Up	2.211	Lipid transport
F1NV02	4355	APOB	Up	3.3	Lipid transport

## Data Availability

The data presented in this study are available in Appendix A.

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
