# Peer review of "Comprehensive Proteome and Acetyl-Proteome Atlas Reveals Hepatic Lipid Metabolism in Layer Hens with Fatty Liver Hemorrhagic Syndrome"

_ijms, 2023, doi:10.3390/ijms24108491_

Round 1

Reviewer 1 Report

Accept in present form. 

Author Response

Dear Reviewer,

Thank you for your kind letter and the constructive comments provided by the reviewers regarding our manuscript (Manuscript ID ijms-2330460). We appreciate your positive feedback and value the suggestions given, which have been instrumental in improving the quality of our article. We have carefully studied the comments and made the necessary corrections to address them.

Please feel free to contact me if you have any further questions or concerns.

Best regards,

Feiruo Huang

Reviewer 2 Report

Review comments on ijms-2330460

Journal:International Journal of Molecular Sciences 

Manuscript ID: ijms-2330460

Type of manuscript:article

Title:Comprehensive Proteome and Acetyl-Proteome Atlas Reveals Hepatic Lipid Metabolism in Layer Hens with Fatty Liver Hemorrhagic Syndrome

Authors:Li Zhang, Enling Wang, Gang Peng, Yi Wang, Feiruo Huang* 

Submitted to section:Molecular Endocrinology and Metabolism,

Major comments:

Previous studies showed that laying hens with fatty liver hemorrhagic syndrome (FLHS) have a phenomenon of hepatic metabolic re-modelling, including lipid accumulation and blockage of lipid transport. A very recent study by Wang et al (ref. #9) on hepatic metabolic changes in the liver of pre-laying and laying hens by analyzing proteome and acetyl-proteome revealed that lysine acetylation modification is a crucial factor in regulating hepatic energy metabolism and lipid synthesis in laying hens. In the present study, Zhang et al. constructed the experimental model of FLHS and conducted a comprehensive hepatic proteome and lysine-acetylation proteome analyses. They found that the up-regulated proteins were primarily associated with fat digestion and absorption, biosynthesis of unsaturated fatty acids, and glycerophospholipid metabolism, while the downregulated proteins were mainly related to bile secretion and amino acid metabolism. Furthermore, the significant acetylated proteins were largely involved in ribosome, fatty acid degradation and PPAR signaling pathway, while the significant deacetylated proteins were related to valine, leucine, and isoleucine degradation.  These results are scientifically very interesting and might be very important for understanding of the pathophysiology of FLHS and developing more effective strategies for its prevention and treatment.

Accordingly, present manuscript is acceptable for publication in “International Journal of Molecular Sciences” if following questions and comments are successfully clarified.

(1)   In the present manuscript, it is stated that “peptide fragments were tested for histone acetylation to ensure consistency with quality control requirements” (Page 5, lines 119~120). Does this mean that the histone acetylation has no role for the formation of FLHS?

(2)   If so, I would like to know how authors can draw such a conclusion. It is well-known that histone acetylation and deacetylation are essential parts of gene regulation.

(3)   Indeed, in Figure 6 (Page 10), “histone modification” and “covalent chromatin modification” (panel B) and “DNA binding” (panel D) are both indicated as one of “biological process” and “molecular function”, respectively, as a results of cluster analysis of differential acetylated proteins. How the authors evaluated these?

Minor comments: 

(1)  In the present manuscript, most of the protein name was shown as abbreviated one. It might be better to give their common name in its first appearance. Otherwise, it is very difficult to understand for most of readers.

(2)  In the Materials and Methods section, TMT labelling was described. However, there was no description what kind of reagents (or kits) were used. Please specify. (Page 18, line 426~428)

Author Response

Responses to reviewers' comments:

Major comments:

Previous studies showed that laying hens with fatty liver hemorrhagic syndrome (FLHS) have a phenomenon of hepatic metabolic re-modelling, including lipid accumulation and blockage of lipid transport. A very recent study by Wang et al (ref. #9) on hepatic metabolic changes in the liver of pre-laying and laying hens by analyzing proteome and acetyl-proteome revealed that lysine acetylation modification is a crucial factor in regulating hepatic energy metabolism and lipid synthesis in laying hens. In the present study, Zhang et al. constructed the experimental model of FLHS and conducted a comprehensive hepatic proteome and lysine-acetylation proteome analyses. They found that the up-regulated proteins were primarily associated with fat digestion and absorption, biosynthesis of unsaturated fatty acids, and glycerophospholipid metabolism, while the downregulated proteins were mainly related to bile secretion and amino acid metabolism. Furthermore, the significant acetylated proteins were largely involved in ribosome, fatty acid degradation and PPAR signaling pathway, while the significant deacetylated proteins were related to valine, leucine, and isoleucine degradation.  These results are scientifically very interesting and might be very important for understanding of the pathophysiology of FLHS and developing more effective strategies for its prevention and treatment.

Accordingly, present manuscript is acceptable for publication in “International Journal of Molecular Sciences” if following questions and comments are successfully clarified.

Response: Thank you very much for giving the manuscript after such a carefully review. Your concerns are very helpful for our manuscript. We have made corresponding changes after careful inspection, which will be detailed explained in the following sections. Besides, revised portion are highlighted by using red colored text in our manuscript.

  • In the present manuscript, it is stated that “peptide fragments were tested for histone acetylation to ensure consistency with quality control requirements” (Page 5, lines 119~120). Does this mean that the histone acetylation has no role for the formation of FLHS?

Response: Thank you for your valuable suggestions after such a carefully review.  We are very sorry for delivering a misleading message and have revised it. The mention of conducting tests on peptide fragments to ensure consistency with quality control requirements does not necessarily imply that histone acetylation has no effect on the formation of FLHS. The identification and clustering analysis of differentially acetylated proteins in FLHS and control samples revealed significant differences in histone acetylation between FLHS and control groups, suggesting that histone acetylation may play a role in the formation of FLHS. Therefore, testing histone acetylation on peptide fragments is only meant to ensure the quality of the experiments, and cannot be used to infer the role of histone acetylation in the formation of FLHS.

Line 124-126:Similarly, peptide fragments were tested to ensure consistency with quality control requirements, with the majority of them ranging from 7 to 20 amino acids in length (Supplementary Figure 1D).

  • If so, I would like to know how authors can draw such a conclusion. It is well-known that histone acetylation and deacetylation are essential parts ofgene regulation.

Response: Thanks for your suggestion which is very helpful in enriching the content of our manuscript. We would like to talk about this issue together with the third point you raised.

  • Indeed, in Figure 6 (Page 10), “histone modification” and “covalent chromatin modification” (panel B) and “DNA binding” (panel D) are both indicated as one of “biological process” and “molecular function”, respectively, as a results of cluster analysis of differential acetylated proteins. How the authors evaluated these?

Response: Thanks for your suggestion which is very helpful in enriching the content of our manuscript. We think it makes sense to explain this issue, so we expand on it in the discussion section to explain.

Line 411-433: As a results of cluster analysis of differential acetylated proteins, histone modifica-tion and covalent chromatin modification and DNA binding are both indicated as one of biological process and molecular function, respectively. Changes in histone modification and covalent chromatin modification can be interpreted as these proteins being involved in the regulation of chromatin state. Histone modification refers to chemical modifications, such as acetylation and methylation, on histone molecules, which can alter the compact-ness of chromatin and thereby affect gene expression [41, 42]. Covalent chromatin modi-fication refers to modifications on non-coding DNA regions that interact with histone molecules, such as methylation and phosphorylation, which can affect chromatin struc-ture and stability, thereby impacting gene expression [43]. Therefore, changes in histone modification and covalent chromatin modification may indicate involvement in the regu-lation of chromatin state and may lead to changes in the expression of certain genes. These clustering results of differentially acetylated proteins in FLHS-affected hens provide a preliminary clue to help researchers explore how these proteins are involved in the reg-ulation of chromatin state and understand their impact on gene expression. Generally, acetylation modification of a protein increases its affinity for DNA, promoting its binding to DNA[44]. This is because acetylation modification can alter the protein's charge state, making it more likely to interact with the negative charges in DNA [45]. In addition, acet-ylation modification can also affect the structure and function of chromatin [46, 47]. By modifying histones with acetylation, the compactness of chromatin can be altered, thereby affecting the accessibility and readability of DNA. Overall, acetylation modification of proteins increases their affinity for DNA, may enhance their function in chromatin, and may affect the accessibility and readability of DNA in hens with FLHS.

Minor comments:

(1) In the present manuscript, most of the protein name was shown as abbreviated one. It might be better to give their common name in its first appearance. Otherwise, it is very difficult to understand for most of readers.

Response: Thanks for your valuable and meaningful suggestions for the manuscript. We have listed the common names of the proteins and highlighted them in red, and also included a summary of the information in the abbreviation table.

Abbreviations 

ACAD8

Acyl-Coa dehydrogenase family member 8 

ACADL

Acyl-Coa dehydrogenase long chain

ACO1

Aconitase 1 

ACO2

Aconitase 2

ACOX1

Acyl-Coa oxidase 1

ACOX2

Acyl-Coa oxidase 2 

ACSBG2

Acyl-Coa synthetase bubblegum family member 2

ACSL1

Acyl-Coa synthetase long-chain family member 1

ACSL5

Acyl-Coa synthetase long chain family member 5

ADH1

Alcohol dehydrogenase 1

ADH2

Alcohol dehydrogenase 2

ADH4

Alcohol dehydrogenase 4

AWAT1 

Acyl-Coa wax alcohol acyltransferase 1

APOAIV

Apolipoprotein A-IV

APOB

Apolipoprotein B

ATP5H

Atp synthase peripheral stalk subunit H

CD36

Cluster of differentiation 36

CPT1

Carnitine palmitoyltransferase 1

CPT1A

Carnitine palmitoyltransferase 1A

CPT2

Carnitine palmitoyltransferase 2

ECI2

Enoyl-Coa delta isomerase 2

EHHADH

Enoyl-Coa hydratase and 3-hydroxyacyl Coa dehydrogenase

ELOVL6

Elongation of very long chain fatty acids protein 6

FABP

Fatty acid binding protein

FABP1

Fatty acid binding protein 1

FABP4

Fatty acid binding protein 4

FABP7

Fatty acid binding protein 7

FADS1

Fatty acid desaturase 1

FADS2

Fatty acid desaturase 2

FAH

Fumarylacetoacetate hydrolase

FGA

Fibrinogen alpha chain

FGB

Fibrinogen beta chain

FGG

Fibrinogen gamma chain

FH

Fumarate hydratase

FLHS

Fatty liver hemorrhagic syndrome

FPR

False positive rate

GO

Gene ontology

GSTM2

Glutathione s-transferase mu 2

HADHA

Hydroxyacyl-Coa dehydrogenase subunit alpha

HAL

Histidine ammonia-lyase

HNF4α

Hepatocyte nuclear factor 4 alpha

HPD

4-Hydroxyphenylpyruvate dioxygenase

HPLC

High-performance liquid chromatography

IDH2

Isocitrate dehydrogenase 2

KEGG

Kyoto encyclopedia of genes and genomes

LAP3

Leucine aminopeptidase 3

MCEE

Methylmalonyl-coa epimerase

MDH1

Malate dehydrogenase 1

MPST

Mercaptopyruvate sulfurtransferase

MTTP

Microsomal triglyceride transfer protein

NAM

N-acetylmuramic acid

NCE

Normalized collision energy

PC

Pyruvate carboxylase

PCYT1A

Phosphate cytidylyltransferase 1 alpha subunit

PDHA1

Pyruvate dehydrogenase e1 component subunit alpha 

PDHA2

Pyruvate dehydrogenase e1 component subunit alpha 2

PLPP1

Phospholipid phosphatase 1

PLIN2

Perilipin 2

PPI

Protein-protein interactions

RPL27A

Ribosomal protein l27a

RPS3A

Ribosomal protein S3a

SDEPs

Significantly differently expressed proteins

SDHA

Succinate dehydrogenase flavoprotein subunit A

SDHB

Succinate dehydrogenase iron-sulfur subunit B

SPE

Solid phase extraction

TSA

Tris-saline-azide

TST

Thiosulfate sulfurtransferase

UROC1

Urocanate hydratase 1

  • In the Materials and Methods section, TMT labelling was described. However, there was no description what kind of reagents (or kits) were used. Please specify. (Page 18, line 426~428)

Response: Thank you for making such a detailed review for the manuscript. We apologize for this error because of our carelessness and have made corresponding changes in the manuscript.

Line 478-480: They were subsequently labeled with the appropriate TMT reagents (Thermo Fisher Scientific) and incubated at 25°C for 2 hours.

Reviewer 3 Report

F. Huang et al reported an article on proteome and acetyl-proteome analysis of laying hens with fatty liver hemorrhagic syndrome which showed that acetylation inhibits hepatic fatty acid oxidation and transport, providing new nutritional regulation options to alleviate the condition. 

The results presented in the paper are interesting and well-supported by the data. However, I recommend the authors revise the discussion section to better contextualize their findings within the existing literature.

Author Response

Responses to reviewers' comments:

  1. Huang et al reported an article on proteome and acetyl-proteome analysis of laying hens with fatty liver hemorrhagic syndrome which showed that acetylation inhibits hepatic fatty acid oxidation and transport, providing new nutritional regulation options to alleviate the condition. 

The results presented in the paper are interesting and well-supported by the data. However, I recommend the authors revise the discussion section to better contextualize their findings within the existing literature.

Response: Thanks for your suggestion which is very helpful in enriching the content of our manuscript. We have revised the discussion section and highlighted it with red color in manuscript.

Line 411-433:As a results of cluster analysis of differential acetylated proteins, histone modifica-tion and covalent chromatin modification and DNA binding are both indicated as one of biological process and molecular function, respectively. Changes in histone modification and covalent chromatin modification can be interpreted as these proteins being involved in the regulation of chromatin state. Histone modification refers to chemical modifications, such as acetylation and methylation, on histone molecules, which can alter the compact-ness of chromatin and thereby affect gene expression [41, 42]. Covalent chromatin modi-fication refers to modifications on non-coding DNA regions that interact with histone molecules, such as methylation and phosphorylation, which can affect chromatin struc-ture and stability, thereby impacting gene expression [43]. Therefore, changes in histone modification and covalent chromatin modification may indicate involvement in the regu-lation of chromatin state and may lead to changes in the expression of certain genes. These clustering results of differentially acetylated proteins in FLHS-affected hens provide a preliminary clue to help researchers explore how these proteins are involved in the reg-ulation of chromatin state and understand their impact on gene expression. Generally, acetylation modification of a protein increases its affinity for DNA, promoting its binding to DNA[44]. This is because acetylation modification can alter the protein's charge state, making it more likely to interact with the negative charges in DNA [45]. In addition, acet-ylation modification can also affect the structure and function of chromatin [46, 47]. By modifying histones with acetylation, the compactness of chromatin can be altered, thereby affecting the accessibility and readability of DNA. Overall, acetylation modification of proteins increases their affinity for DNA, may enhance their function in chromatin, and may affect the accessibility and readability of DNA in hens with FLHS.

Reviewer 4 Report

Fatty liver hemorrhage syndrome (FLHS) is a nutritional metabolic disease occurring in laying hens at the peak of egg production, which seriously decreases production perfrmance and egg production rate of laying hens. While the mechanism remains unclear. In this article, the author conducted a comprehensive hepatic proteome and acetyl-proteome analysis in both normal and FLHS-affaected hens, found that acetylation inhibited hepatic fatty acid oxidation and transport in FLHS-affected hens. These findings enriched our understanding about the pathogenesis of FLHS. And it also provides new nutritional regulation options to alleviate FLHS. Thus, this article is suitable to be published in IJMS.

Author Response

(The authors gave the same response as above.)

Round 2

Reviewer 2 Report

Review comments on ijms-2330460-peer-review-v2

Journal: International Journal of Molecular Sciences 

Manuscript ID: ijms-2330460-peer-review-v2

Type of manuscript: article

Title: Comprehensive Proteome and Acetyl-Proteome Atlas Reveals Hepatic Lipid Metabolism in Layer Hens with Fatty Liver Hemorrhagic Syndrome

Authors: Li Zhang, Enling Wang, Gang Peng, Yi Wang, Feiruo Huang* 

Submitted to section: Molecular Endocrinology and Metabolism,

Major comments:

(1)   In the revised version of the manuscript, authors made several revisions and additions in light of the suggestions and comments raised by reviewers including me. Most of the responses and answers in the cover letter were very appropriate and reasonable.

(2)   Although authors added a new discussion section on “histone modification and covalent chromatin modification and DNA binding” (page 18, lines 411-433) in response to reviewer’s comment, authors should be more careful about this.  Actually this newly-added discussion part was rather abrupt and peculiar, since there were no descriptions at all on these protein modifications in Result section, i.e., in section 2.6 and in section 2.7. These words appear only in panels of Figure 6.  Therefore, in Result section, authors should mention their observations on histone modification and covalent chromatin modification that made the above discussion part being possible and meaningful. 

(3)   Accordingly, in the Abstract section, it might be better to add the authors’ important conclusion, such as “acetylation modifications predominantly affect protein activity rather than expression, consistent with its role as a post-translational modification” (Page 18, lines 435-435) to make the newly-added discussion part meaningful.

(4)   Therefore, the revised manuscript can be acceptable for publication in “International Journal of Molecular Sciences” after these additional and appropriate modifications.

Author Response

Dear Reviewer,

Thank you for your valuable feedback and the constructive comments provided by the reviewers regarding our manuscript (Manuscript ID ijms-2330460). We greatly appreciate your suggestions, which have been crucial in enhancing the quality of our article. We have thoroughly reviewed your comments and made the necessary revisions to address them. We are committed to providing accurate and informative content and strive to continuously improve our work. Thank you again for your input.

Please feel free to contact me if you have any further questions or concerns.

Best regards,

Feiruo Huang

Responses to reviewers' comments:

Major comments:

(1)   In the revised version of the manuscript, authors made several revisions and additions in light of the suggestions and comments raised by reviewers including me.  Most of the responses and answers in the cover letter were very appropriate and reasonable.

Response: Thank you for taking the time to review our manuscript and for your valuable feedback. We greatly appreciate your comments and suggestions, and we are pleased to hear that you found our revisions and additions to be appropriate and reasonable. We have carefully considered all the comments raised by the reviewers, including yours, and we believe that these suggestions have helped to strengthen the manuscript. We hope that the revised version of the manuscript is now suitable for publication in the journal. Thank you again for your time and assistance in the review process.

(2)   Although authors added a new discussion section on “histone modification and covalent chromatin modification and DNA binding” (page 18, lines 411-433) in response to reviewer’s comment, authors should be more careful about this.   Actually this newly-added discussion part was rather abrupt and peculiar, since there were no descriptions at all on these protein modifications in Result section, i.e., in section 2.6 and in section 2.7.  These words appear only in panels of Figure 6.   Therefore, in Result section, authors should mention their observations on histone modification and covalent chromatin modification that made the above discussion part being possible and meaningful.

Response:  Thank you for your feedback and for bringing this to our attention. We appreciate your careful attention to our manuscript and agree that we could have provided more context regarding the histone modification and covalent chromatin modification discussed in the newly-added discussion section. We will carefully revise the Results section to include a more detailed description of our observations on these protein modifications, and how they relate to the findings presented in the discussion section. We apologize for any confusion that this may have caused, and we thank you for helping us to improve the clarity and coherence of our manuscript. Thank you again for your time and effort in reviewing our manuscript.

Line 201-204: Regarding biological processes (Figure 6B), the PwSDALs in the Q1 category were mainly involved in triglyceride metabolism, mitochondrial gene expression, protein targeting to the endoplasmic reticulum, covalent chromatin modification, and histone modification.

Line 221-224: The PwSDALs in Q2 category were associated with fatty acid oxidation-related biological functions, such as 3-hydroxy acyl-CoA dehydrogenase activity, long chain-enoyl-CoA hydratase activity, and mitochondrial function such as succinate dehydrogenase activity, electron transfer activity, and DNA binding.

Line 250-253: Proteins enriched in cluster 1 were involved in ribosomal functions, with RPS27A, a key enzyme involved in ribosome structure formation, transcriptional regulation, DNA binding and protein synthesis, showing the highest degree of connectivity.

Line 430-432: In our research, RPS27A showing the highest degree of connectivity in PPI network. Studies have shown that RPS27A can bind to specific DNA sequences and affect DNA replication and repair processes [46]

(3)  Accordingly, in the Abstract section, it might be better to add the authors’ important conclusion, such as “acetylation modifications predominantly affect protein activity rather than expression, consistent with its role as a post-translational modification” (Page 18, lines 435-435) to make the newly-added discussion part meaningful.

Response: Thank you for your valuable feedback regarding our manuscript. We agree that including the conclusion regarding the effect of acetylation on protein activity rather than expression would enhance the significance of the newly-added discussion section. We have carefully reviewed and revised the Abstract section of our manuscript to incorporate this important conclusion. The revised Abstract section now includes the sentence you suggested. We appreciate your suggestion and believe that this modification will improve the clarity and impact of our manuscript. Thank you again for your feedback and for your consideration of our manuscript for publication.

Line 18-20: Overall, these results demonstrate that acetylation inhibits hepatic fatty acid oxidation and transport in hens with FLHS, and mainly exerts its effects by affecting protein activity rather than expression. 

(4)  Therefore, the revised manuscript can be acceptable for publication in “International Journal of Molecular Sciences” after these additional and appropriate modifications.

Response: Thank you very much for your positive feedback on our revised manuscript. We greatly appreciate your time and effort in reviewing the manuscript and providing valuable feedback that helped us improve the quality of our research. We have carefully considered your comments and suggestions and have made the necessary modifications to address the issues raised. We are pleased to hear that the revised manuscript can be acceptable for publication in "International Journal of Molecular Sciences". Once again, we thank you for your valuable feedback and for considering our manuscript for publication.

Round 3

Reviewer 2 Report

Review comments on ijms-2330460-peer-review-v3

Journal: International Journal of Molecular Sciences 

Manuscript ID: ijms-2330460-peer-review-v3

Type of manuscript: article

Title: Comprehensive Proteome and Acetyl-Proteome Atlas Reveals Hepatic Lipid Metabolism in Layer Hens with Fatty Liver Hemorrhagic Syndrome

Authors: Li Zhang, Enling Wang, Gang Peng, Yi Wang, Feiruo Huang* 

Submitted to section: Molecular Endocrinology and Metabolism,

Major comments:

(1)   In re-revised version of the manuscript, authors made several revisions and additions in light of the suggestions and comments raised in my previous review comments. All of the revisions and additions including in Abstract were very appropriate and reasonable.

(2)   Accordingly, this re-revised manuscript can be acceptable for publication in “International Journal of Molecular Sciences” in the present form.
